# Potent NTD-Targeting Neutralizing Antibodies against SARS-CoV-2 Selected from a Synthetic Immune System

**DOI:** 10.3390/vaccines11040771

**Published:** 2023-03-31

**Authors:** Wenping Li, Fulian Wang, Yu Li, Lei Yan, Lili Liu, Wei Zhu, Peixiang Ma, Xiaojie Shi, Guang Yang

**Affiliations:** 1Shanghai Institute for Advanced Immunochemical Studies, ShanghaiTech University, Shanghai 201210, China; liwp@shanghaitech.edu.cn (W.L.);; 2School of Life Science and Technology, ShanghaiTech University, Shanghai 201210, China; 3CAS Center for Excellence in Molecular Cell Science, Shanghai Institute of Biochemistry and Cell Biology, Shanghai 200031, China; 4University of Chinese Academy of Sciences, Beijing 101408, China; 5Shanghai Key Laboratory of Orthopedic Implants, Department of Orthopedic Surgery, Shanghai Ninth People’s Hospital, Shanghai Jiao Tong University School of Medicine, Shanghai 200031, China

**Keywords:** synthetic immune system, neutralizing antibody, combinatorial antibody library, somatic hypermutation, SARS-CoV-2

## Abstract

The majority of neutralizing antibodies (NAbs) against SARS-CoV-2 recognize the receptor-binding domain (RBD) of the spike (S) protein. As an escaping strategy, the RBD of the virus is highly variable, evolving mutations to thwart a natural immune response or vaccination. Targeting non-RBD regions of the S protein thus provides a viable alternative to generating potential, robust NAbs. Using a pre-pandemic combinatorial antibody library of 10^11^, through an alternate negative and positive screening strategy, 11 non-RBD-targeting antibodies are identified. Amongst one NAb that binds specifically to the N-terminal domain of the S protein, SA3, shows mutually non-exclusive binding of the angiotensin-converting enzyme 2 receptor with the S protein. SA3 appears to be insensitive to the conformational change and to interact with both the “open” and “closed” configurations of the trimeric S protein. SA3 shows compatible neutralization as S-E6, an RBD-targeting NAb, against the wild type and variant of concern (VOC) B.1.351 (Beta) of the SARS-CoV-2 pseudo virus. More importantly, the combination of SA3 with S-E6 is synergistic and recovers from the 10-fold loss in neutralization efficacy against the VOC B.1.351 pseudo virus.

## 1. Introduction

SARS-CoV-2 has infected over 700 million people and caused over 6 million deaths as of March 2023 (https://covid19.who.int/ (accessed on 8 March 2023)), which poses a big challenge to the global public health system [1]. In the global fight against coronaviruses, neutralizing monoclonal antibodies provide not only potential treatment but also valuable epitopes for vaccine development. Most antibody studies focused on the coronavirus spike (S) protein, a key structural viral protein for its recognition and membrane fusion to host cells [2]. This is critical to the coronavirus in determining host infectivity and transmission capacity. Generally, the S protein is divided into two subunits: the S1 subunit responsible for receptor binding and the S2 subunit responsible for cell membrane fusion [3]. The S1 subunit consists of a receptor-binding domain (RBD) that binds to the receptor protein of host cells, angiotensin-converting enzyme 2 (ACE2) [4], and a N-terminal domain (NTD) [2,5]. The gene fragment encoding RBD of the S protein is the most variable portion of the coronavirus genome, generating a series of variants of concern (VOC), i.e., Beta, Delta, and Omicron [6,7]. Monoclonal antibodies targeting the S protein with strong neutralizing activity against SARS-CoV-2 are powerful therapeutic agents in clinical interventions [8]. Most neutralizing antibodies (NAbs) isolated from convalescent patients bind to the RBD of the S protein rather than the NTD [9,10,11,12]. However, due to its high genetic variability, targeting RBD inevitably results in a high degree of immune escape of the SARS-CoV-2 virus [6,13,14].

The NTD of the S1 subunit, on the other hand, is functionally less susceptible to host cell interaction and thus less immunogenic. As a consequence, the NTD less frequently induces NAbs with high affinity from host immune responses [9,10,11,12]. Recent studies demonstrated several convalescent serum-isolated monoclonal antibodies targeting NTD were capable of neutralizing SARS-CoV-2 viral infection in vitro [12,15,16]. Interestingly, most NTD-targeting SARS-CoV-2 NAbs did not interfere with the binding of the S protein to ACE2. In addition to the intrinsic anti-viral activity, targeting non-RBD epitopes such as NTD could generate orthogonal pairs of antibodies specific to the same coronavirus suitable for cocktail therapy.

The combinatorial antibody library (CAL) technology is a method to reconstruct the antibody diversity repertoires of the individual B cell immune system in a test tube by DNA rearrangement in vitro [17,18]. The CAL as a synthetic immune system has advantages, such as the super-high diversity over other approaches in selecting NAbs. CAL includes genetic material from memory cells, providing a record of all of the antibodies that populations have made, irrespective of whether they are currently being produced, was termed “fossil record” by us [19]. This enabled us to rapidly respond to pandemics like COVID-19 and discover NAbs with high affinity from the history memory of human beings. In current study, we designed a selection scheme that took full advantage of the approach of CAL to identify human antibodies that recognize non-RBD epitopes of SARS-CoV-2 S protein. To enhance the immunogenic response for the non-RBD epitopes, a negative screening step against the recombinant RBD was applied. A pre-pandemic CAL containing 10^11^ naive human single-chain variable fragment (scFv) antibodies [20] was then enriched by three rounds of alternate negative/positive selection against the recombinant RBD and S protein of SARS-CoV-2, respectively. The antibody, SA3, recognizing a novel epitope on the NTD of the SARS-CoV-2 S protein was identified with more somatic hypermutations compared with NTD-targeting NAbs from convalescent COVID-19 patients [12,15,16], which manifested in the greatly improved neutralization activity against the SARS-CoV-2 pseudo virus. More importantly, SA3 showed potent neutralization activity alone or in combination with an RBD-targeting NAb, S-E6 [19], against a pseudo virus derived from an immune escape variant of SARS-CoV-2.

## 2. Materials and Methods

### 2.1. Cell Culture

The HEK293-F cell line (ThermoFisher Scientific, #R79007, Waltham, MA, USA) was maintained in the Freestyle 293 expression medium (#12338026, Thermo, Life Technologies Corporation, Grand Island, NY, USA). HEK293T cells were obtained from ATCC and maintained in DMEM (Gibco, #C11995500BT, ThermoFisher Biochemical Products, Beijing, China) supplemented with 10% FBS (Gemini, #900108, Punta Mita, Mexico). All the cells were cultured at 37 °C with 5% CO_2_.

### 2.2. Expression and Purification of Antigen and Antibody

The extracellular domain of SARS-CoV-2 spike (GI 43740568, amino acid 1-1208) with T4 fibritin (T4F) motif and 6-histidine tag (S-trimer-His) and SARS-CoV-2 spike RBD (amino acids 319-541, S-RBD) with 6-histidine tag or human Fc (*h*Fc) tag (S-RBD-His or S-RBD-*h*Fc) were cloned into the pFuse expression vectors (#pfuse-hg1fc2, InvivoGen USA, San Diego, CA, USA). Then they were transiently expressed in HEK293F cells and purified by Mabselect columns (#17-5199-01, GE Healthcare Bio-Science AB, Uppsala, Sweden) or HisTrap Excel columns (Cytiva, #17371205, GE Healthcare Bio-Science AB, Uppsala, Sweden) according to the manufacturer’s instructions. As reported by Wrapp et al., the extracellular domain of the SARS-CoV-2 spike sequence was modified to get the prefusion conformation as follows: the furin cleavage site (residues 682–685) was mutated to GSAS, KV (residues 986-987) was mutated to PP, and. a HRV 3C cleavage site was inserted before the 6-histidine tag or *h*Fc tag [21]. Combinatorial antibodies in the scFv-Fc format or full-length human IgG1 construct in this same pFuse expression vector were expressed in HEK293F cells for 4 days and then purified by Mabselect chromatography according to the manufacturer’s instructions. Purified recombinant antigens and antibodies were buffer exchanged into the PBS buffer (pH 7.4) using centrifugal concentrators (Amicon Ultra-4, Merck Millipore, #UFC803096, Tullagreen, Carrigtwohill, Ireland).

### 2.3. Phage Panning

SARS-CoV-2 spike non-RBD region-specific scFv antibodies were selected from a combinatorial human scFv antibody library (10^11^ members) displayed on phages after three rounds of affinity enrichment. In each round, the phage particles were first incubated with biotinylated S-RBD-*h*Fc immobilized on the Streptavidin (SA)-coated magnetic beads (#11206D, Invitrogen, Thermo Fisher Scientific Baltics UAB, Vilnius, Norway) for the negative selection. Phages that did not bind to S-RBD-*h*Fc were collected and further incubated with biotinylated S-trimer-His immobilized on the SA-coated magnetic beads for positive selection. Antigen-bound phages were eluted with Glycine-HCl (pH 2.2) after each round of screening. XL1-Blue cells were used to amplify the output phages for the next round of panning. After three iterations, colonies were picked and analyzed by phage ELISA. All positive clones were sequenced using Sanger sequencing.

### 2.4. ELISA

ELISA procedures were described previously [19]. Phage ELISA was used to select positive clones after panning. Briefly, antigens at a concentration of 2 ng μL^−1^ were coated on 384-well-plates (20 μL per well) at 4 °C overnight. After blocking and PBST washing, the antigen coated plates were incubated with 50 μL XL1-Blue culture supernatants containing phages in each well at room temperature (RT) for 1 h. Anti-M13 bacteriophage antibody conjugated with HRP (20 μL per well, dilution factor 1:5000; #11973-MM05T-H, Sino Biological, Beijing, China) and the ABTS solution (20 μL per well, Roche, #11684302001) were used as the secondary antibody and the detection substrate, respectively, following PBST washes. Absorbance at 405 nm was measured (Enspire, PerkinElmer, Singapore).

To determine the binding properties of antibodies to antigens, briefly, the antibodies or antigens were coated on 96-well plates (2 ng μL^−1^ in 100 μL of PBS buffer per well). Biotinylated antigen or antibody solutions with 1:5 serial dilution in PBS buffer from 100 to 0.0064 nM were incubated in the coated wells (100 μL per well) at RT for 1 h. Streptavidin conjugated with HRP (ThermoFisher Scientific, #21130) was used as the secondary detector. The coloration of the substrate and absorbance detection are the same as described above.

For selected antibodies to the S protein and the S-RBD, the recombinant antibodies were coated. Biotinylated S-trimer-His and S-RBD-His solutions were added to the pre-coated wells.

For SA3-*h*Fc to the S-NTD (GI 43740568, amino acid 14-290), the S-NTD-His protein (#DRA45, Novoprotein, Shanghai, China) was coated on a nickel-coated plate (ThermoFisher Scientific, #15442). Biotinylated SA3-*h*Fc solutions were incubated in the pre-coated wells.

For the competition assay, SA3-*h*Fc or S-E6-IgG4 were coated. 2 nM biotinylated S-trimer-His was pre-incubated with of various concentrations ACE2-*m*Fc or SA3-*h*Fc at RT for 1 h. Then the mixture (100 μL) was added to the pre-coated wells and incubated for another 1 h at RT.

For comparison of the “open” and “closed” configurations of the S protein, SA3-*h*Fc was coated. Biotinylated S-trimer-His with or without 50 μM Linoleic Acid (LA) pre-incubated were incubated in the pre-coated wells.

### 2.5. Affinity Determination by Biolayer Interferometry (BLI)

Binding affinities were performed by BLI on the Octet RED96 (Molecular Devices LLC, San Jose, CA, USA). Biotinylated SA3-*h*Fc was loaded onto a SA biosensor (#18-5019, Sartorius, ForteBio, PALL Life Sciences, Shanghai, China) at 5 μg mL^−1^ in PBST-B buffer (PBS containing 0.02% Tween-20 and 0.05% BSA). The SA-bio-SA3 sensor was dipped into PBST-B for 60 sec to establish a baseline, then incubated in antigen solutions of various concentrations (1:2 serial dilution from 200 to 6.25 nM) to record the progression curves of association, followed by dissociation progression in a PBST-B buffer. The S-trimer-His antigens with and without 50 μM LA pre-incubation was compared in the conformation sensitive detection.

Sensor regeneration and equilibration, R*_max_*, *k_on_*, *k_off_* and K_D_ fitting were carried out as previously described.

### 2.6. Antibody Recognition of Cell Surface-Displayed Spikes by FACS

The spike proteins of SARS-CoV-2 wild-type (GI 43740568, amino acids 1-1273), SARS-CoV (GI 1489668, amino acids 1-1255), MERS-CoV (GI 14254594, amino acids 1-1353), HCoV-229E (GI 918758, amino acids 1-1173), HCoV-HKU1 (GI 3200426, amino acids 1-1351), HCoV-NL63 (GI 2943499, amino acids 1-1356), HCoV-OC43 (GI 39105218, amino acids 1-1353), or alanine-scan mutants of SARS-CoV-2 wild-type spike were C-terminally fused with EGFP with a P2A self-cleaving peptide inserted between, were cloned into pcDNA3.1 vectors (Invitrogen, #V79020) and transiently expressed on HEK293T cells. The cells were collected and incubated with 100 nM testing antibodies for 30 min at 4 °C in FACS buffer (PBS, 0.05% BSA, and 2 mM EDTA). The resulting cells were washed twice with the FACS buffer, followed by incubating in a staining solution containing an Alexa555 conjugated secondary antibody (1:800 dilution, #A21433, Invitrogen, Life Technologies Corporation, Eugene, OR, USA) at 4 °C for 30 min. Finally, the cells were washed twice, re-suspended, and then analyzed on the flow cytometer (CytoFLEX S, Beckman Coulter Life Sciences, Suzhou, China).

### 2.7. Size-Exclusion-High-Performance Liquid Chromatography

20 μL antibody solution in a PBS buffer (pH 7.4, 0.5 μg μL^−1^) was applied to an Agilent Bio SEC-5, 500A HPLC system using PBS buffer (pH 7.4) as the mobile phase at a flow rate of 0.5 mL min^−1^. The percentage of aggregation and degradant compositions of a testing antibody was monitored by absorbance change at wavelength 280 nm at the corresponding retention times.

### 2.8. Pseudovirus-Based Neutralization Assay

Pseudo virus (PSV)-based neutralization assay was described previously [19]. Briefly, PSV of SARS-CoV-2 wild-type or mutant spike Δ19 (19 amino acids truncated at the C-terminus) with mCherry and Luciferase were first produced in HEK293T cells according to the reference. Then, HEK293T cells expressing hACE2 were seeded into a 96-well, white-opaque plate at a density of 1 × 10^4^ per well. Testing antibodies serially diluted in DMEM with 10% FBS (dilution factor: 3.16, from 200 nm to 6.3 fM) were incubated with an equal volume of PSV at 37 °C for 30 min. No PSV or PSV in the absence of antibodies were set as controls for normalization. 100 µL of the mixtures were transferred to the HEK293T/hACE2 cells. Fresh media was changed 16 h after treatment for an additional culture at 48 h. PSV transduction was evaluated by luciferase activity using the Bright-Lumi Firefly Luciferase Reporter Gene Assay Kit (Beyotime, #RG015M) according to the manufacturer’s instructions. Data fitting was carried out in GraphPad Prism 8.3. The combination index (CI) in the PSV neutralization assay was calculated by the CompuSyn program to evaluate the synergism according to the program instruction. CI > 1 indicates antagonism, CI = 1 indicates additive effect, and CI < 1 indicates synergism.

### 2.9. Chemical Crosslinking and Mass Spectrometry

S-trimer-His and SA3-*h*Fc were crosslinked using collision-induced dissociation (CID)-cleavable cross-linker, disuccinimidyl sulfoxide (DSSO) following the described procedure [22]. Briefly, S-trimer-His and SA3-*h*Fc were mixed in PBS and incubated for 30 min at RT. Crosslinking was performed for 30 min at RT after adding DSSO (ThermoFisher Scientific, #A33545) to the mixture with 100-fold molar excess and quenching it with excess Tris (1 M, pH 8.0) for 10 min at RT. Then the crosslinked products were digested with chymotrypsin. The LC-MS^n^ data of digested peptides were collected on the Orbitrap Fusion Tribrid (ThermoFisher Scientific) with an on-line NanoLC system and analyzed using the CID-MS^2^-MS^3^ strategy as previously described [23].

### 2.10. Quantification and Statistical Analysis

All values in the text and figures are presented as the mean ± SEM of independent experiments with given sizes (*n*). Graphs were compiled, and statistical analyses were performed with Prism software (GraphPad). Statistical significance was evaluated with the two-tailed unpaired *t*-test when comparing two groups and with the one-way analysis of variance (ANOVA) when comparing more than two samples. Other statistical details are indicated in the figures and legends.

## 3. Results

### 3.1. Selection of Antibodies Targeting the Non-RBD Regions of SARS-CoV-2 Spike Protein

The biotinylated recombinant SARS-CoV-2 spike RBD fused with human Fc (S-RBD-*h*Fc) and the full-length spike extracellular domain with T4F motif and C-terminal His-tag (S-trimer-His) immobilized on streptavidin (SA)-coated magnetic beads were panned against a pre-pandemic combinatorial scFv antibody phage library containing 10^11^ members generated from peripheral blood mononuclear cells of 50 healthy donors in 1999 [20]. In order to select non-RBD-binding antibodies of interest, an alternate negative/positive panning strategy was developed for a total of three rounds of panning (Figure 1a). The binding signal ratios of S-trimer-His to S-RBD-*h*Fc phagemids dramatically rose to 3- and 16-folds in the second and third rounds of the process (Figure 1b). Positive phages from round 3 were examined using phage ELISA against S-RBD-*h*Fc and S-trimer-His. Eleven unique clones were identified by Sanger sequencing and re-confirmed to bind S-trimer-His but not S-RBD-*h*Fc (Figure 1c,d). The 11 enriched positive clonal sequences were shown to harbor multiple somatic hypermutations (SHMs) and derive from multiple different germlines (Appendix A). Three clones, SA3, SA4, and SB4, derived from the same germlines (IGHV1-18 and IGKV1-39) showed 11 identical SHMs in the heavy chain and various SHMs (0, 4, 6, respectively) in the light chain. Purified SA3, SA4, and SB4 fused with *m*Fc were characterized and validated to bind S-Trimer-His and spike proteins expressed on the cell surface (Appendix A).

### 3.2. Kinetic Characterization and Mode of Interaction for Non-RBD Targeting Antibody SA3

SA3 was first converted into the formats of scFv-linked human Fc-tag (SA3-*h*Fc) and full-length IgG1 (SA3-IgG1). Both formats had good thermostability at 4 °C or room temperature (RT) over a period of 7 days’ incubation (Appendix A). Similar to that of SA3-*m*Fc, the *h*Fc and IgG1 forms of SA3 were shown to bind to the S-trimer-His with an apparent EC_50_ value of 0.15 ± 0.02 (Figure 2a) and 4.4 ± 0.5 nM (Figure 2b), respectively, but not the RBD domain. SA3-*h*Fc compared to the full length SA3-IgG1 showed at least one order of magnitude more potent binding and was used in the following experiments.

SA3-*h*Fc appeared to display a “fast-on, slow-off” kinetics to S-trimer-His with an apparent dissociation constant (K_D_) value of 0.15 ± 0.01 nM determined by biolayer interferometry (BLI) using a 1:1 fitting model (Figure 2c). Similarly, BLI failed to detect any interaction between SA3-*h*Fc and S-RBD-His.

The binding of SA3-*h*Fc to S-trimer-His in the presence or absence of Linoleic Acid (LA) which was previously shown to bind the RBD of SARS-CoV-2 S protein tightly (K_D_ = 41.1 ± 16 nM) and lock the S protein in the “closed” conformation to reduce interaction with ACE2 in vitro [24] was carried out by BLI and showed no measurable difference in binding affinities (Figure 2c). This implies strongly a distinct binding feature of SA3 that is independent of the interaction between ACE2 and S protein. Consistent with this notion, in a competitive binding experiment of SA3-*h*Fc, ACE2-*m*Fc, and S-trimer-His, SA3-*h*Fc and ACE2-*m*Fc showed mutually non-exclusive binding to S-trimer-His (Figure 2d).

We ectopically expressed spike proteins of all seven coronaviruses that infect humans [5,25] including SARS-CoV-2, SARS-CoV, HCoV-229E, HCoV-OC43, HCoV-NL63, HCoV-HKU1, and MERS-CoV on the surface of HEK293T cells and tested the cross-reactivity of SA3 to them. FACS analyses showed that SA3 specifically bound to the S protein of SARS-CoV-2 but not the other six coronaviruses (Figure 2e).

### 3.3. Epitope Mapping of SA3

To dissect the mode of interaction between the SARS-CoV-2 S protein and SA3, we sought to determine the binding epitopes of SA3 on the S protein using high-resolution mass spectrometry (MS) and mutagenesis. The S-trimer-His and SA3-*h*Fc were cross-linked and digested, followed by “shot-gun” tandem MS analyses. Integrative analyses of CID-induced cleavage of interlinked peptides in MS/MS^2^ and MS^3^ of single peptide chain fragment ions revealed high crosslinking scores (Figure 3a) on residues K187 of QGNF[K]NLR, K458 of [K]SNLKPFER, K417 of QIAPGQTG[K]IADYNYK, and K206 of IYS[K]HTPINLVR peptide from the S protein (Figure 3b and Appendix A). These peptides locate in the RBD and NTD of the S protein, where the light chain of SA3 appears to interact with both the NTD and RBD, and the heavy chain interacts with the RBD (Figure 3a–c). To further validate the NTD-targeting of SA3, the binding of SA3-*h*Fc to S-NTD-His was examined by ELISA, which showed an apparent EC_50_ value of 16 ± 2 nM (Figure 3d).

To identify the critical epitope residues involved in SA3 binding, we performed FACS-based alanine substitutions at the peptide regions of Spike_185–189_, Spike_204–208_, Spike_415–419_, and Spike_456–460_. Comparing with the NTD-targeting antibody 4A8 [16], Spike_185–189_ on the N4 loop of the S protein was found to be the main interacting site for SA3-*h*Fc (Figure 3e). Each individual residue on Spike_185-189_ was next mutated to alanine to identify the critical residues. As shown in Figure 3f, point mutation in F186A abolished the binding of SA3-*h*Fc to S proteins, while K187A and L189A showed partially disrupted binding of SA3-*h*Fc. The N4 loop at the NTD of the S protein, therefore, most likely consists of the epitope residues for SA3.

### 3.4. Mutually Non-Exclusive Binding and Synergistic Neutralization of SA3 and S-E6 against SARS-CoV-2

The mutually non-exclusive binding of SA3-*h*Fc vs. ACE2-*m*Fc to the S protein (Figure 2d) observed above suggests SA3 is most likely orthogonal to an RBD-binding antibody with the S protein. S-E6 was previously identified as an RBD-targeting NAb through competitive blockage of the ACE2 attachment and binding of viral spikes [19]. The neutralization activities of the NTD-targeting antibody SA3-*h*Fc against wild-type (WT) and variant SARS-CoV-2 spikes were evaluated in a PSV infection assay and compared to S-E6-IgG4, which showed compatible potent neutralization against SARS-CoV-2 WT and variants in a dose-dependent manner. The apparent NT_50_ values of SA3-*h*Fc against WT, Beta, and Delta were determined to be 0.063 ± 0.011, 0.23 ± 0.05, and 1.5 ± 0.2 nM, respectively (Figure 4a). For Omicron (BA.1), a significant decrease in SA3 neutralization potency occurred with the dramatic remodeling of the NTD surface of the Omicron [26]. However, it still maintained over 90% inhibition of efficacy at high concentrations (1 μM).

To validate whether the binding of S-E6 and SA3 to the S protein is orthogonal, competitive binding to S-trimer-His was tested, which showed mutually non-exclusive binding (Figure 4b). PSV assay was next applied in the neutralization of Beta variant of SARS-CoV-2. As expected, combination of S-E6 and SA3 showed synergistic neutralization activity with paired antibodies displaying more potent neutralization than each alone (Figure 4c). The combination index (CI) value of the 1:1 orthogonal antibody mixture at NT_50_ was determined to be 0.53 indicating a nearly 50% enhancement in viral neutralization.

## 4. Discussion

Epitope mapping studies on SARS-CoV-2 NAbs revealed several sites on the S protein that appeared with high frequency in host immune responses [9,15,27,28]. The majority of the highly immunogenic sites are located in the RBD of the S protein since the RBD is highly exposed on the viral membrane and attachment site of the virus to the host cell [9,10,11,15]. However, as a defense strategy of viruses, RBD is highly susceptible to mutations through genetic drift and host immune adaptation. Here we present an approach beyond human immune responses in vivo to select antibodies targeting the non-RBD sites on the S protein using the CAL. The natural CAL is a synthetic immune system that consists of not only an infinite diversity of antibody sequences but also “fossil records” of the immune responses of each individual donor within the library [17,18,19,22,29]. In our previous study of SARS-CoV-2, multiple potent RBD-targeting antibodies with strong neutralizing activities were discovered from the CAL constructed before the COVID-19 pandemic [19]. Further in the current study, we demonstrated that epitopes of weak immunogenicity in the natural immune system could be sensitized and enhanced to elicit strong immune responses in a synthetic immune system.

The negative screening step helped to eliminate interference from RBD-binding antibodies. A total of 11 non-RBD targeting antibodies were identified, representing four different germlines for heavy chains (IGHV5-51, 3-30, 1-18, and 1-69). Amongst SA3, specific binding to the N4 loop in NTD outside the supersite was observed [15,16,28] with high affinity binding to spike (K_D_ = 0.15 ± 0.01 nM) and strong neutralizing activity. Six convalescent non-RBD targeting antibodies from the CoV-AbDab database that share the same germline of heavy and light chains (IGHV1-18/IGKV1-39) with SA3 showed no neutralization efficacy against SARS-CoV-2 [30,31,32,33] (Appendix A). Detailed sequence alignment indicates that SA3 contains 11 SHMs, compared to much lower levels of SHMs in those from convalescent patients. In acute infections (during the first week of COVID-19 infection), most antibodies undergo minimal SHM with limited clonal expansion [34]. The previous observation of patterns of convergence in potent NAb lineages across different donors [35] and the lack of neutralization activity for the convalescent non-RBD targeting antibodies strongly implicate the necessity of repeated immune stimulation for weak immunogenic epitopes. This illustrates well the value of CAL, with its “fossil record” of providing antibodies to weak immunogens.

Comparing to the published NTD-targeting neutralizing antibodies isolated from convalescent patients with typical IC_50_ values inhibiting SARS-CoV-2 infection between 2 and 700 ng/mL (about 10 nM to 5 μM) [12,15,30,36,37], SA3 screened from the in vitro synthetic immune system showed more potent neutralizing activity with an IC_50_ value of 0.063 ± 0.011 nM against SARS-CoV-2 pseudo virus. SA3 and the RBD-targeting antibody S-E6 were also shown to bind to the S protein in a mutually non-exclusive manner, which makes them an ideal orthogonal pair. Indeed, SA3 and S-E6 showed a remarkable additive effect in the pseudoviral neutralization assay of the SARS-CoV-2 VOC B.1.351 variant. The selection strategy of orthogonal antibodies with weak immunogenicity presented here facilitates the development of new therapeutic antibodies, detection methods, and preventative vaccines to meet the challenge of present and future pandemics. For instance, in view of the reductions in response to SARS-CoV-2 vaccines in patients with autoimmune diseases such as systemic sclerosis and immunosuppressant treatment, more vaccine doses or vaccination prior to planned immunosuppression is recommended [38]. Therefore, additional therapeutic antibody treatment, especially for those with less immunogenic epitopes, might be beneficial to the populations to compensate for the deficiency.

## Figures and Tables

**Figure 1 vaccines-11-00771-f001:**
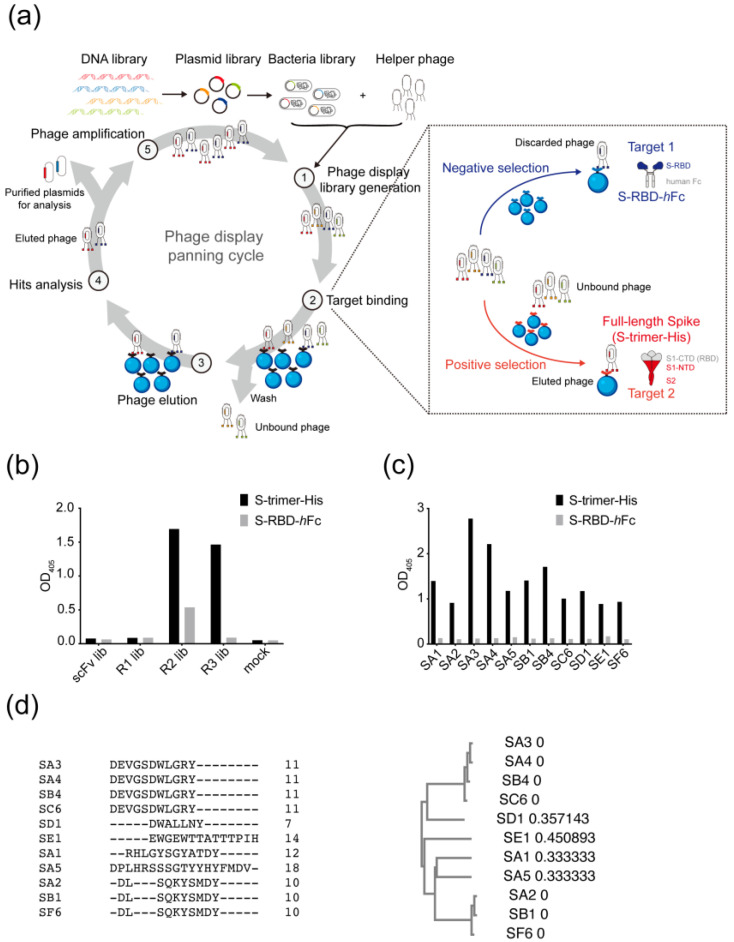
Combinatorial antibody library screening for non-RBD targeting antibodies against SARS-CoV-2. (**a**) Schematic representation of a 5-step phage panning cycle (①–⑤). The insertion box describes a detailed alternate negative (highlighted in blue)/positive (highlighted in red) selection strategy for step ② of the Target binding. (**b**) Phage ELISA results of three rounds of panning against S-RBD-*h*Fc (grey) and S-trimer-His (black). (**c**) Phage ELISA results of enriched 11 unique clones of round 3 against S-RBD-*h*Fc (grey) and S-trimer-His (black). (**d**) Amino acid sequence alignment of CDR-H3 of the enriched antibodies by multiple sequence alignment in CLUSTAL O (1.2.4). The numbers besides the clone names in the right panel represent the branch lengths, indicating the evolutionary distances between two consecutive nodes in the phylogenetic tree.

**Figure 2 vaccines-11-00771-f002:**
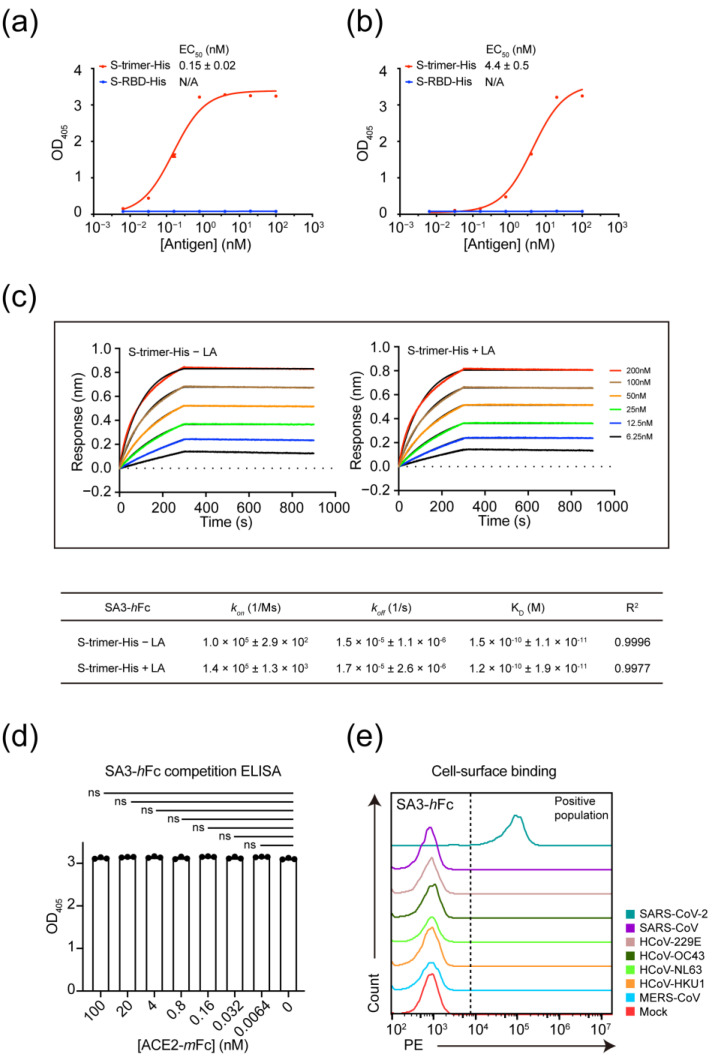
Characterization of interactions between SA3 and S proteins of coronaviruses. (**a**) Comparison of interactions between SA3-*h*Fc and biotinylated S-RBD-His (blue traces) and S-trimer-His (red traces) by ELISA. (**b**) Comparison of interactions between SA3-IgG1 and biotinylated S-RBD-His (blue traces) and S-trimer-His (red traces) by ELISA. (**c**) Kinetic characterization of SA3-*h*Fc binding to S-trimer-His by BLI in the absence (left panel) and presence (right panel) of 50 μM LA (Linoleic Acid). Sensorgram curves at various concentrations (colored traces) were fitted to the superimposed black lines by a 1:1 binding model with a globally linked R*_max_* using the ForteBio. Kinetic parameters are summarized in the table. (**d**) Competitive binding of SA3-*h*Fc to S-trimer-His in the presence of various concentrations of ACE2-*m*Fc by ELISA. (**e**) FACS analyses of the association of SA3-*h*Fc with HEK293T cells expressing spikes of various coronaviruses (colored traces indicated in the figure) on the membrane. All the ELISA experiments were carried out in triplicate (*n* = 3) and monitored by the absorbance change at wavelength 405 nm (OD_405_). Dose-response curves were fitted by the Hill equation to obtain EC_50_ values. Statistical analysis was carried out by one-way ANOVA in GraphPad Prism.

**Figure 3 vaccines-11-00771-f003:**
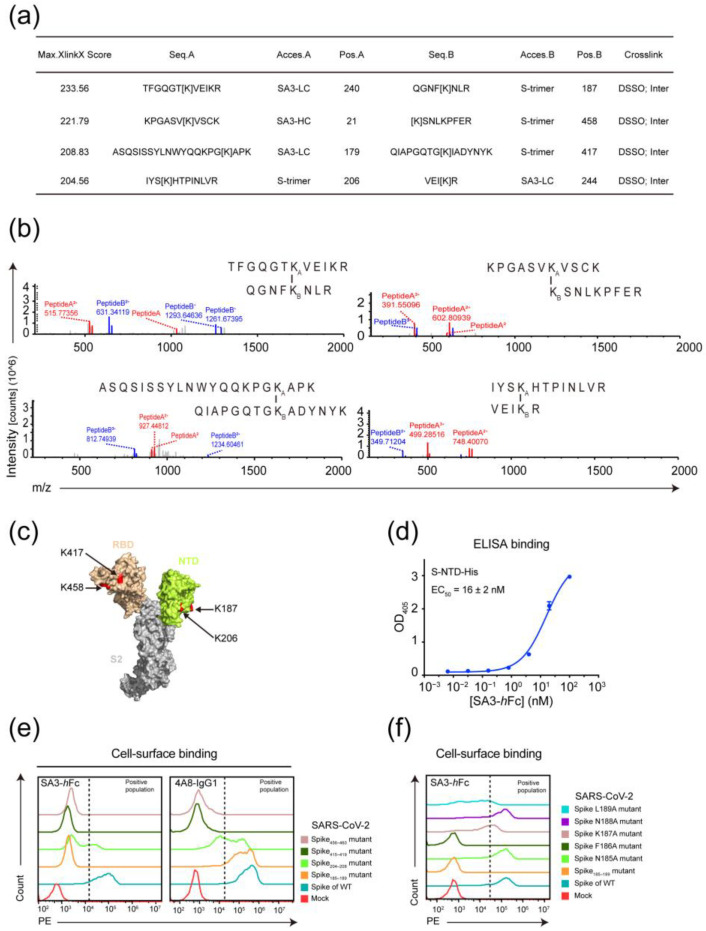
Epitope mapping of SA3 on the spike of SARS-CoV-2. (**a**) Top 4 hit peptides with XlinkX scores in high-resolution MS analyses. (**b**) Secondary MS showing crosslinked peptides. (**c**) Location of SA3 binding sites on the spike of SARS-CoV-2. The monomer structure of the SARS-CoV-2 spike is displayed by PyMOL. The RBD is shown in orange, and the NTD is shown in lemon yellow. The crosslinked lysines mediated by SA3 binding on the spike are shown in red and indicated by arrows. (**d**) Dose-dependent binding of SA3-*h*Fc to S-NTD-His was monitored in triplicates as OD_405_ changed by ELISA assay and fitted to obtain the apparent EC_50_ value. (**e**) FACS-based cassette mutagenesis of 5 residues on Spike_185–189_, Spike_204–208_, Spike_415–419_, and Spike_456–460_ of SARS-CoV-2 using alanine-scan. (**f**) FACS-based point mutagenesis on Spike_185–189_ of SARS-CoV-2 using alanine-scan. Colored traces represent the association of SA3-*h*Fc or 4A8-IgG1 with different mutant or WT spikes of SARS-CoV-2 on HEK293T cells membrane in (**e**,**f**). Positive binding populations were as indicated.

**Figure 4 vaccines-11-00771-f004:**
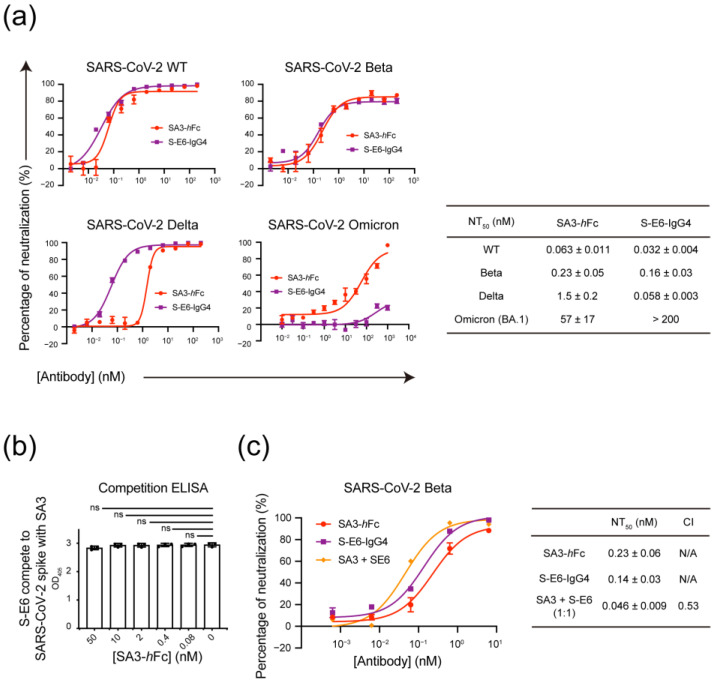
Mutually non-exclusive binding and synergistic neutralization activities of SA3 and S-E6. (**a**) Dose-dependent neutralization of SA3-*h*Fc and S-E6-IgG4 against WT and variant PSVs. Apparent NT_50_ values are summarized in the table. (**b**) Competitive binding of S-E6-IgG4 to S-trimer-His in the presence of indicated concentrations of SA3-*h*Fc by ELISA. OD_405_ values were measured and plotted in triplicates (*n* = 3). (**c**) Percentage neutralization of S-E6-IgG4 and SA3-*h*Fc either alone or in a 1:1 combination against SARS-CoV-2 Beta PSV at various testing concentrations. The combination index (CI) value at NT_50_ was calculated using the CompuSyn program. CI < 1 is defined as synergism. All the dose-response curves were measured in triplicates (*n* = 3). The apparent NT_50_ values were determined by Hill analyses using one-way ANOVA in GraphPad Prism.

## Data Availability

Appendix A is available online. Raw data is available from the corresponding authors upon reasonable request.

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
