# Peer review of "Potent NTD-Targeting Neutralizing Antibodies against SARS-CoV-2 Selected from a Synthetic Immune System"

_vaccines, 2023, doi:10.3390/vaccines11040771_

Round 1

Reviewer 1 Report

  • Major comments: 

In this study, Wenping Li et al. identified several non-RBD targeting antibodies through alternate negative and positive screening strategies from a pre-pandemic combinatorial antibody library. Among them, SA3 could bind specifically to the N-terminal domain of the SARS-CoV-2 S protein without the interruption of the binding of the angiotensin-converting enzyme 2 receptor with the S protein. Besides, SA3 shows compatible neutralization as S-E6, an RBD-targeting NAb, against the wild type and variant of concern (VOC) B.1.351 of SARS-CoV-2 pseudovirus. More importantly, the authors also showed that these two antibodies could be used synergistically.

The study is relevant to the field and well-organized.

  • General concept comments

Here are some considerations for the study:

1.      The antiviral effects of SA3 against Omicron pseudovirus should be tested, as the Omicron variant is the most prevalent one, and wildtype and Beta variant are outdated in the circumstances of nowadays.

2.       It would be better to include the structure data or structure docking of SA3 complex with SARS-CoV-2 S protein.

  • Specific comments:

1)      Line 22, suggests changing “SARS-CoV-2” to “SARS-CoV-2 pseudovirus” since only pseudovirus assay was tested in the study. The same suggestion also applied to Line 24.

2)      Line 48-49, while the RBD of SARS-CoV-2 is subject to high genetic variability, the NTD domain is not so “conserved” either, as manifested by the frequent mutations in the circulating variants of SARS-CoV-2.

3)      Please provide more information about the combinatorial antibody library (CAL) in the section of the introduction.

4)      Line 76, the HEK293-F cell line, is maintained in 5% CO2?

5)      Line 77, section 2.2. Expression and Purification of Antigen and Antibody. The specific location and accession number of the SARS-CoV-2 spike and RBD should be labeled. Besides, no NTD protein was purified here. What are the source and specific locations of NTD protein? Besides, the sequence information of the Fc tag and full-length human IgG1 should be listed as well.

6)      Line 106, suggests adding “respectively” between “substrate” and “following”.

7)      Line 107, the ELISA reactions were not stopped and instead, measured at 405 nm?

8)      Line 131, what does “PBST-B” mean?

9)      Lines 139-140, other HCoVs, and their specific accession numbers should be listed here. Besides, please detail the P2A-EGFP tag here.

10)  Please specify what the numbers mean in the right panel of Figure 1(d).

11)  Lines 257-258, since SA3-hFc could bind with the S-trimer-His with high affinity (“fast-on, slow-off,” is it still necessary to cross-link them for the MS analyses?

12)  Lines 269-270, according to Figure 3e, other mutations seemed also abolish the interaction of SA3-hFc to the S-trimer? Why not map other epitopes of SA3? Is F186 the only amino acid responsible for the binding with the S-trimer?

13)  Lines 305-306, what does 1∶1 orthogonal antibody mixture mean? Is it the molar ratio or the mass ratio? Did you test other ratios other than 1:1 for better CI?

14)  Line 338, the lack of neutralization efficacy to SARS-CoV-2 of the 6 convalescent non-RBD targeting antibodies needs a reference to support it.

Author Response

General concept comments

Here are some considerations for the study:

1) The antiviral effects of SA3 against Omicron pseudovirus should be tested, as the Omicron variant is the most prevalent one, and wildtype and Beta variant are outdated in the circumstances of nowadays.

Response: Thanks for the suggestion. We did evaluate the scope of neutralization activity of the SA3 antibody selected by combinatorial antibody library approach. We found it bind to and neutralize against different variants including Beta, Delta, and Omicron (BA.1) in the pseudovirus infection assay but with significantly reduced potency for Delta and Omicron. The NT50 values for Delta and Omicron are 1.5 ± 0.2 nM and 57 ± 17 nM, respectively. We updated the Figure 4 and Results Section 3.4 accordingly in the manuscript.

2) It would be better to include the structure data or structure docking of SA3 complex with SARS-CoV-2 S protein.

Response: Since the crystal structure of SA3 is not available, we are doing the docking using molecular dynamics (MD) simulation. It will take a couple of weeks of computation to complete data collection.

Specific comments:

1) Line 22, suggests changing “SARS-CoV-2” to “SARS-CoV-2 pseudovirus” since only pseudovirus assay was tested in the study. The same suggestion also applied to Line 24.

Response: We have revised the manuscript accordingly.

2) Line 48-49, while the RBD of SARS-CoV-2 is subject to high genetic variability, the NTD domain is not so “conserved” either, as manifested by the frequent mutations in the circulating variants of SARS-CoV-2.

Response: Thank you for the suggestion. We have removed the statement of “genetically more conserved” in the revised manuscript now.

3) Please provide more information about the combinatorial antibody library (CAL) in the section of the introduction.

Response: As recommended, more detailed description of combinatorial antibody library (CAL) has been added in Introduction section of the revised manuscript.

4) Line 76, the HEK293-F cell line, is maintained in 5% CO2?

Response: Yes, the culture condition is 5% CO2 for the HEK293F cells.

5) Line 77, section 2.2. Expression and Purification of Antigen and Antibody. The specific location and accession number of the SARS-CoV-2 spike and RBD should be labeled. Besides, no NTD protein was purified here. What are the source and specific locations of NTD protein? Besides, the sequence information of the Fc tag and full-length human IgG1 should be listed as well.

Response: We have added these details in Sections 2.2 and 2.4 of the revised manuscript correspondingly.

6) Line 106, suggests adding “respectively” between “substrate” and “following”.

Response: We have revised the manuscript accordingly of the revised manuscript.

7) Line 107, the ELISA reactions were not stopped and instead, measured at 405 nm?

Response: Yes. According to the manufacturer’s protocol, absorbance changes of ABTS at 405 nm (green color) could be monitored continuously.

8) Line 131, what does “PBST-B” mean?

Response: PBST-B represents a PBS buffer containing 0.02% Tween-20 and 0.05% BSA, which was described at its first appearance in the manuscript.

9) Lines 139-140, other HCoVs, and their specific accession numbers should be listed here. Besides, please detail the P2A-EGFP tag here.

Response: We have added the sequence information and details of P2A-EGFP in Materials and Methods, Section 2.6 of the revised manuscript accordingly. P2A is a self-cleaving peptide that generates polyproteins within a same open reading frame during the protein translation process.

10) Please specify what the numbers mean in the right panel of Figure 1(d).

Response: The numbers are the real branch lengths representing the evolutionary distances between two consecutive nodes. We have added this information in the figure legends in the revised manuscript.

11) Lines 257-258, since SA3-hFc could bind with the S-trimer-His with high affinity (“fast-on, slow-off,” is it still necessary to cross-link them for the MS analyses?

Response: Peptide mapping experiment using LC-MS/MS analyses requires covalent cross linking of the peptide ligand and antibody protein.

12) Lines 269-270, according to Figure 3e, other mutations seemed also abolish the interaction of SA3-hFc to the S-trimer? Why not map other epitopes of SA3? Is F186 the only amino acid responsible for the binding with the S-trimer?

Response: In alanine-scan study, Spike 415-419 and Spike 456-460 are located in the S-RBD region. Cassette mutations at these two sequences were shown to abolish the interaction of both SA3 and 4A8 antibodies to S protein. Since the 4A8 antibody is known to target specifically the S-NTD region by cryo-EM [1], the observed loss of binding affinity most likely a consequence of an indirect conformational change.

13) Lines 305-306, what does 1∶1 orthogonal antibody mixture mean? Is it the molar ratio or the mass ratio? Did you test other ratios other than 1:1 for better CI?

Response: It means the molar ratio. Since the NT50 values of the two antibodies on beta variant were close, we arbitrarily chose a 1:1 molar ratio to unravel the possible synergism. We did not optimize the molar ratio to get the best inhibitory effect.

14) Line 338, the lack of neutralization efficacy to SARS-CoV-2 of the 6 convalescent non-RBD targeting antibodies needs a reference to support it.

Response: We have inserted the supporting references correspondingly.

References

  1. Chi, X.; Yan, R.; Zhang, J.; Zhang, G.; Zhang, Y.; Hao, M.; Zhang, Z.; Fan, P.; Dong, Y.; Yang, Y.; et al. A neutralizing human antibody binds to the N-terminal domain of the Spike protein of SARS-CoV-2. Science 2020, 369, 650-655, doi:10.1126/science.abc6952.

Reviewer 2 Report

The authors screened combinatorial antibody library to identify human NAbs that recognize non-RBD epitopes of Spike protein and obtained 3 non-RBD targeting antibodies. The manuscript is well-written and SA3 antibody looks promising.

Comments for the authors:

Major points:

1.    P.2, Line 91: Please explain “SA-coated”. I could not find #21925 by Google search.

2.    Please make sure all the catalogue numbers are correct.

3.    P.5, Line 201: Please explain how the authors selected 3 clones. What was the antigen used for dot blot?

Minor points:

1.    P.1, Line 22: Please add (Beta) after B.1.351

2.    P.2, Line 79: “6-histone tag” should be “6-histidine tag”.

3.    P.3, Line 112: “Sino Biological, #11973-MM05T-H” is anti-M13 antibody. Please confirm.

4.    P.3, Line 118: “Thermo, #QK220689” should be “#15442”.

Author Response

Comments and Suggestions for Authors

The authors screened combinatorial antibody library to identify human NAbs that recognize non-RBD epitopes of Spike protein and obtained 3 non-RBD targeting antibodies. The manuscript is well-written and SA3 antibody looks promising.

Comments for the authors:

Major points:

1) P.2, Line 91: Please explain “SA-coated”. I could not find #21925 by Google search.

Response: Thanks for pointing out the typo in the text. “SA-coated” means “Streptavidin-coated magnetic beads” from Invitrogen catalog # 11206D. This information is updated in Materials and Methods, Section 2.3.

2) Please make sure all the catalogue numbers are correct.

Response: We have proofread the catalogue number information for all the related commercial reagents.

3) P.5, Line 201: Please explain how the authors selected 3 clones. What was the antigen used for dot blot?

Response: The 11 clones with unique antibody sequences were cloned into the pFUSE vector (#pFUSE-mIgG1-Fc2, Invivogen) in the scFv-mouse Fc format followed by expression in HEK293F cells. The antibody-containing cell culture supernatants were dotted on the membrane and hybridized by the anti-mouse antibody conjugated with HRP (Abcam, #ab97265) to check the expression of these antibodies. Finally, the well-expressed clone SA3, SA4, and SB4 were selected.

Minor points:

1) P.1, Line 22: Please add (Beta) after B.1.351

Response: We have revised the content accordingly.

2) P.2, Line 79: “6-histone tag” should be “6-histidine tag”.

Response: We have revised the content accordingly.

3) P.3, Line 112: “Sino Biological, #11973-MM05T-H” is anti-M13 antibody. Please confirm.

Response: It should be “Thermo, #21130”. We have revised the content accordingly.

4) P.3, Line 118: “Thermo, #QK220689” should be “#15442”.

Response: We have revised the content accordingly.

Reviewer 3 Report

1) what is the main differences of the antibodies produced by the authors and those previously published.

2) Which is the novelty of the antibodies produced by the authors?. 

3) Evaluation of the neutralization capacity of the Ab against Omicron, should be part of the main figures and text. Not in supplementary material. The authors should focus their experiments with Omicron virus. For example, figure 4 need to be changed to evaluated Omicron not Beta (and Beta sent it su supplemental material).

4) Line 197, chance "11" by "Eleven". 

5) In discussion the authors did not comparte their antibodies against other published, advantages, disadvantages, etc., deeply. 

Author Response

Comments and Suggestions for Authors

1) what is the main differences of the antibodies produced by the authors and those previously published.

Response: The combinatorial antibody library approach, an in vitro synthetic immune system, consists of a much larger diversity (at least 3 orders of magnitude) comparing to the natural immune system, which makes selection of rare and highly specific antibodies possible. Secondly, as we demonstrated in the current study, in vitro selection process could be customized to enrich antibodies targeting less immunogenic epitopes. Furthermore, since the library not only capture the diversity but also the history of immune responses of donors’ B-cell repertoire. This enabled us to rapidly identify mature and highly specific monoclonal antibodies targeting SARS-CoV-2 in the COVID-19 pandemic, which are in marked difference comparing to those discovered from the convalescent patients of SARS-CoV-2.

2) Which is the novelty of the antibodies produced by the authors?

Response: As discussed above, the synthetic immune system could produce rare and highly specific antibodies that the natural immune system may not be able generate due to the immune tolerance issue in the body.   

3) Evaluation of the neutralization capacity of the Ab against Omicron, should be part of the main figures and text. Not in supplementary material. The authors should focus their experiments with Omicron virus. For example, figure 4 need to be changed to evaluated Omicron not Beta (and Beta sent it to supplemental material).

Response: Please see the response to the first reviewer. We have made modifications and supplements with the corresponding content.

4) Line 197, chance "11" by "Eleven".

Response: We’ve changed “11” to “Eleven” in Section 3.1 as suggested.

5) In discussion the authors did not compare their antibodies against other published, advantages, disadvantages, etc., deeply.

Response: Comparing to the published NTD-targeting antibodies isolated from convalescent patients with typical IC50 values inhibiting SARS-CoV-2 infection between 2 to 700 ng/mL (about from 10 nM to 5 μM), or complete non-neutralization [2-6], SA3 screened from the in vitro synthetic immune system showed more potent neutralizing activity with IC50 value of 0.063 ± 0.011 nM against SARS-CoV-2 pseudovirus. We have added this accordingly in the discussion section.

References

  1. Brouwer, P.J.M.; Caniels, T.G.; van der Straten, K.; Snitselaar, J.L.; Aldon, Y.; Bangaru, S.; Torres, J.L.; Okba, N.M.A.; Claireaux, M.; Kerster, G.; et al. Potent neutralizing antibodies from COVID-19 patients define multiple targets of vulnerability. Science 2020, 369, 643-650, doi:10.1126/science.abc5902.
  2. Liu, L.; Wang, P.; Nair, M.S.; Yu, J.; Rapp, M.; Wang, Q.; Luo, Y.; Chan, J.F.; Sahi, V.; Figueroa, A.; et al. Potent neutralizing antibodies against multiple epitopes on SARS-CoV-2 spike. Nature 2020, 584, 450-456, doi:10.1038/s41586-020-2571-7.
  3. McCallum, M.; De Marco, A.; Lempp, F.A.; Tortorici, M.A.; Pinto, D.; Walls, A.C.; Beltramello, M.; Chen, A.; Liu, Z.; Zatta, F.; et al. N-terminal domain antigenic mapping reveals a site of vulnerability for SARS-CoV-2. Cell 2021, 184, 2332-2347 e2316, doi:10.1016/j.cell.2021.03.028.
  4. Suryadevara, N.; Shrihari, S.; Gilchuk, P.; VanBlargan, L.A.; Binshtein, E.; Zost, S.J.; Nargi, R.S.; Sutton, R.E.; Winkler, E.S.; Chen, E.C.; et al. Neutralizing and protective human monoclonal antibodies recognizing the N-terminal domain of the SARS-CoV-2 spike protein. Cell 2021, 184, 2316-2331 e2315, doi:10.1016/j.cell.2021.03.029.
  5. Zost, S.J.; Gilchuk, P.; Chen, R.E.; Case, J.B.; Reidy, J.X.; Trivette, A.; Nargi, R.S.; Sutton, R.E.; Suryadevara, N.; Chen, E.C.; et al. Rapid isolation and profiling of a diverse panel of human monoclonal antibodies targeting the SARS-CoV-2 spike protein. Nat Med 2020, 26, 1422-1427, doi:10.1038/s41591-020-0998-x.

Reviewer 4 Report

The paper is interesting and well written. The identification of a serological test for the analysis of antibodies against Sars-Cov2 particularly nautralizing versus spike is of key importance above all in patients with chronic diseases including autoimmune diseases. For this reason, considering as references papers by Murdaca et al concerning vaccination in patients with systemic sclerosis, I suggest to discuss if the detection of antibodies may be decrease by the chronic disease itself and/or immunosuppressant agents including biologics. I underline the indication of vaccine these patients 3-4 weeks before starting DMARDs or bDMARDs (see and add as reference paper by Murdaca et al as position statement for vaccination in systemic sclerosis published in vaccines)

Author Response

Comments and Suggestions for Authors

The paper is interesting and well written. The identification of a serological test for the analysis of antibodies against Sars-Cov2 particularly nautralizing versus spike is of key importance above all in patients with chronic diseases including autoimmune diseases. For this reason, considering as references papers by Murdaca et al concerning vaccination in patients with systemic sclerosis, I suggest to discuss if the detection of antibodies may be decrease by the chronic disease itself and/or immunosuppressant agents including biologics. I underline the indication of vaccine these patients 3-4 weeks before starting DMARDs or bDMARDs (see and add as reference paper by Murdaca et al as position statement for vaccination in systemic sclerosis published in vaccines)

Response: Thanks for reviewer’s comment on vaccination in populations with specific autoimmune diseases. We have included corresponding discussion in the revised manuscript and cited the relevant reference of Murdaca et al published in Vaccines, 2021.

Round 2

Reviewer 1 Report

I think that the manuscript has been improved, and the authors have addressed most of my concerns.

Here are two minor points that are needed to be addressed:

1)    Since the crystal structure of SA3 is not available, the molecular docking result using molecular dynamics (MD) simulation is expected.

2)    What are the source and specific locations of NTD protein used in Figure 3(d)?

Reviewer 3 Report

A improvement was detected